# **Identifying Orographic Gravity Waves in 3D Observations via Backward Ray Tracing**

Sebastian Rhode<sup>1</sup>

<sup>1</sup>Forschungszentrum Jülich, Institute of Climate and Energy Systems: Stratosphere (ICE-4), Germany

**Correspondence:** Sebastian Rhode (s.rhode@fz-juelich.de)

**Abstract.** Atmospheric gravity waves (GWs) generated by orography, commonly referred to as mountain waves (MWs), play a key role in driving atmospheric circulation and in modulating phenomena such as sudden stratospheric warmings (SSWs). Their contribution, however, is difficult to disentangle from the full spectrum of observed or simulated GWs. Here, we present a methodology to isolate the MW component of GW observations by combining simulated infrared limb imager measurements with backward ray tracing. This approach enables a systematic separation of GW momentum flux (GWMF) carried by MWs from the total observed signal.

As a case study, we analyze the 2018/19 Northern Hemisphere New Year SSW period, presenting global distributions and time series of GWMF partitioned into orographic and residual components. The ensemble of backward ray trajectories shows a strong correspondence between inferred MW sources and surface topography, supporting the robustness of the method. On average, the identified MWs account for only a minor fraction of the observed GWMF, but they can dominate episodically, including prior to the onset of the SSW.

These results highlight the potential of combining satellite observations with ray tracing to achieve source attribution of GWs. The method's effectiveness depends on the accuracy of retrieved GW parameters, and we therefore include a sensitivity analysis of parameter uncertainties in the appendix.

## 15 1 Introduction

Gravity waves (GWs) are one of the major processes of momentum transport within the middle atmosphere. As they propagate from their source, the GWs carry momentum (GW momentum flux, GWMF) along their path both in the vertical and horizontal direction. When GWs dissipate or break due to, e.g., instability or critical level filtering, they deposit this momentum into the mean flow and accelerate or decelerate the large-scale wind field. This forcing is one of the main drivers of atmospheric circulation and interacts with global-scale dynamics such as sudden stratospheric warming events (SSW) via preconditioning the polar vortex before the breakdown unfolds.

The prediction and forecast of these dynamic phenomena in climate models is highly dependent on simplified parametrizations because the small scales of GWs are, in general, not well resolved (e.g. Martinez-Andradas et al., 2025). These parametrizations have their own limitations such as not representing the horizontal propagation of GWs, simplified breaking mechanisms, e.g., saturation schemes that do not capture complex nonlinear interactions, and the lack of secondary wave generation and

other in-situ wave sources. These limitations are caused by both technical limitations (e.g., computational resolution) as well as by incomplete observational understanding of the GW life cycle on global scale. Furthermore, the parametrizations are usually separated by source processes, orographic and non-orographic GWs, and tuned empirically such that a realistic circulation emerges in the middle atmosphere. Direct, source-separated observations of GWMF could greatly advance parameterizations by enabling process-based tuning due to the clear separation of drag sources.

The observational challenge: The observations needed for a better process understanding of GW interactions and the general GW activity in the middle atmosphere require high data quality and the capability to determine wave parameters, such as the 3D wave vector, for individual GWs. Such high-resolution datasets can be obtained, for example, from 3D temperature measurements by airglow sounders, especially when combined with radar and lidar observations (Reichert et al., 2019, 2024), or from satellite observations of, e.g., the AIRS (Atmospheric Infrared Sounder) instrument (Wright et al., 2017; Ern et al., 2017; Hindley et al., 2019; Ern et al., 2022; Noble et al., 2024). The current observations are, however, limited in their usefulness for improving GW parameterizations: ground-based measurements fail to represent the global distribution and AIRS observations suffer from coarse vertical resolution, leading to missing a large part of the GW spectrum (e.g. Preusse et al., 2008; Alexander et al., 2010; Meyer et al., 2018).

Previous studies have demonstrated that a limb-viewing infrared imager could be the way forward to resolve a broad range of wave scales globally (Ungermann et al., 2010; Preusse et al., 2014; Rhode et al., 2024). Currently, there are two technically mature approaches: an imaging Fourier transform spectrometer as proposed for ESA Earth Explorer 11 candidate CAIRT and a vertically-scanning grating spectrometer proposed for NASA Earth System Explorer candidate STRIVE. A satellite instrument based on either approach could be the first mission that allows for GW observations of sufficiently high quality throughout the middle atmosphere for improving our process understanding.

Missing source attribution: To leverage observations for improving the representation of GWs in global-scale climate models, the observed GWs need to be unambiguously linked to their sources. Such source attribution usually relies on collocating the measurements with potential source regions at lower altitudes directly below. In this study, we will show for orographic GWs that this is not reliable. If all wave parameters can be determined reliably (as in the case of CAIRT, Rhode et al., 2024), ray tracing can be employed for the calculation of trajectories, which give a better picture of the origin of the GWs. Preusse et al. (2014) used a similar approach and showed that convective GWs could propagate farther than 10 000 km from their origin in the middle atmosphere. This methodology alleviates the uncertainty of the horizontal propagation of the observed GWs, offering a more robust source process linking than direct collocation.

In the case of mountain waves such source identification was demonstrated from limb-imager observations from research aircraft (Krisch et al., 2017, 2020; Krasauskas et al., 2023). Yet, a systematic method to reliably separate orographic from non-orographic sources on part-global scale using observations alone does not exist. This study aims at closing this gap by presenting a methodology using ray tracing to identify orographic GW sources in high resolution satellite observations. In particular, we employ the ray tracer GROGRAT (Marks and Eckermann, 1995) and investigate simulated CAIRT observations (Rhode et al., 2024). The following section gives an overview of the used simulated observations and auxiliary data. Sec. 3 details the source attribution criteria and methodology before results for the period of 2018-12-10 to 2019-03-29 are shown

in Sec. 4 and 5 for individual situations and ensemble analysis, respectively. A summary and concluding remarks are given in Sec. 6.

#### 2 Data

The simulated atmospheric data used in this study is based on the ECMWF IFS (ECMWF, 2023), which is run at  $T_{CO}1279$  resolution (corresponding to about 9 km horizontal resolution) and provided on  $0.25^{\circ}$  grid spacing. The investigated period is the time of the 2018/19 northern hemispheric New Year SSW and covers 2018-12-10 through 2019-03-31 with the central date being 2018-12-31.

The data is scale separated following Strube et al. (2020): A zonal Fourier transformation is applied and the spectrum truncated beyond ZWN 7. Afterwards, to preserve east-west oriented GWs, the spectra are smoothed in latitude and altitude using a Savitzky-Golay filter of about 10° width in latitude and 3 km in altitude (both using 3rd order polynomials). The rather low cutoff at ZWN 7 is used, as this would also be an applicable cutoff for CAIRT observations (Mathew et al., 2025). The inverse Fourier transformation of the so-processed spectra gives the atmospheric background. This background will be used to separate the GW residuals from the synthetic observations of temperature.

The synthetic observations are generated from this data set by sampling the temperature of single snapshots (here 12:00 UTC of each day is used) to a satellite observation grid with a sampling of 50 km along-track, 25 km across-track and 1 km in the vertical and by performing an end-to-end simulation. This consists of a forward-calculation of synthetic radiances and a subsequent retrieval using the JURASSIC2 software, which was used in similar studies in the past (e.g. Hoffmann, 2006; Ungermann et al., 2015; Krasauskas et al., 2021). White noise according to CAIRT instrument specification is added to the synthetic radiances and the retrieval leads to realistic noise structures of the temperature observations, which therefore give a good representation of the actual observation performance (see also Rhode et al., 2024).

GW parameters are determined in the same way as done by Rhode et al. (2024), i.e., using a limited-volume sinusoidal fit approach (S3D, Lehmann et al., 2012). This gives a full description of the wave vector and amplitude of up to 4 superposed wave modes per location. Here, we focus on the observations at a typical stratospheric altitude of 35 km. For the purpose of ray tracing, all identified GWs are assumed to propagate upwards, which is a good approximation for the considered altitudes (Guest et al., 2000; de la Torre et al., 2023).

Furthermore, the ray tracing requires a background atmosphere through which the GWs propagate. For this, we use ERA5 wind and temperature data (Hersbach et al., 2018) instead of the high resolution IFS. The data is scale separated in the same way as the high resolution data above for consistency. The large-scale background of temperatures and winds give the ray tracing background atmosphere. This preprocessing is done 6 hourly to get a time-dependent background atmosphere. These fields are slowly changing in horizontal direction by design, hence, the horizontal resolution can be reduced to 1° in latitude and longitude.

**Figure 1.** Flow chart depicting the steps for the MW identification via ray tracing. All waves are initialized for backward ray tracing in a given background atmosphere. Afterwards, a selection is generated from the ray tracing and the observed phase speeds. The selection then gives the MW part of the observed GWs.

The ray tracer used in this study is GROGRAT (first introduced by Marks and Eckermann, 1995) in the updated and modified version described by Rhode et al. (2023). It internally interpolates the background atmosphere to the GW location via cubic splines. All GWs characterized by S3D are initialized for backward ray tracing.

Ray tracing is intrinsically sensitive to the wave parameters of individual wave events. Therefore, a brief sensitivity study of how uncertainties in the wave parameters, and hence the spectral analysis, affect the backward ray tracing is given in App. A. The main result is that the wavelengths and wave direction should be known to about 10% and 10°, respectively. The horizontal wavelength, however, is less critical than the vertical wavelength.

#### 3 Mountain wave identification

The identification of MWs used in this study is based on the phase speed and on the backward ray tracing trajectories: MWs are often characterized by their low ground-based phase speed. However, other sources of GWs can also generate GWs with low phase speeds, leading to this criterion being not sufficient for the attribution to orographic sources. For increased confidence, we analyze the backward ray tracing trajectory of each observed GW. The waves are are expected to propagate far down close to the surface in the vicinity of high orographic elevation. The topology data for surface elevation is taken from the NOAA ETOPO global relief model (Amante and Eakins, 2009; NOAA National Centers for Environmental Information, 2022).

A general overview of the MW identification process is shown in Fig. 1. To summarize, the identification of the GWs as MWs requires the following criteria to be fulfilled:

- 1. Ground-based phase speed  $c_{ph,qb} 




Applying these filtering criteria to the observed GW data allows to separate the observations into MWs and non-orographic GWs. Note that this selection can be fully automatized and could, in principle, be applied to large sets of model or observation data (and for satellite observations as a same day product).

The backward trajectories of all GWs can furthermore be used to analyse the propagation properties and distances of the observed GWs and the subset of MWs in particular. For instance, many MWs travel quite far from their supposed initial source, as will be seen in the next section.

## 4 Observation separation - case study

As a first case study, we investigate the regional distribution of MWs before and after the 2018/19 SSW. Figure 2 shows two-week-aggregated simulated observations of all GWMF and the isolated orographic GWMF for the period before and way past the 2018/19 SSW after the vortex has already recovered. The distribution of all observed GWMF completely differs between both periods: before the SSW, strong GW activity in the Mongolia region dominates, while afterwards, most orographic GWMF is located above the North Atlantic. This pattern is even more enhanced if only the orographic part of the observations is considered: the GWMF is mostly located above Mongolia during the pre-SSW period, while the North Atlantic and northern Canada completely dominate in terms of GWMF after the vortex recovered.

The GWs that are attributed to orographic origin here, however, contribute a comparatively small part of the total observed GWMF. This is also seen in the daily timeline of GWMF in Fig. 3. The MW activity is very intermittent across the considered period, which is expected from previous studies (e.g., Jiang et al., 2002; Plougonven et al., 2013; Kuchar et al., 2020). This intermittency is further enhanced by the limited sampling of the satellite: For MWs to be seen, both the wind conditions need to be favorable and the satellite needs to overpass the excited MWs. Combined, this can potentially lead to an underrepresentation of MWs in the measurements. Less intermittent sources, on the other hand, are sampled regularly and generally well represented by the observations. Nevertheless, the strongest GW events in the observations are mostly driven by orographic GWMF. Overall the MWs account for around 15% of the GWMF on average, which can rise to ~40% on individual days.

## 5 Source attribution

We further analyze the orographic part of the GW observations for the full period of 2018-12-10 to 2019-03-31. This gives insight in the validity of the MW selection criteria and a check of the often-made assumptions of vertical-only propagation for MWs. The latter is achieved by investigating the propagation behaviour of the identified MWs. In order to have the full statistics and global coverage the data of the whole period is aggregated and we are not limiting the analysis to a specific case.

# 5.1 GWMF distribution

The distribution of orographic GWMF at observation altitude of 35 km and at the corresponding endpoint of the ray traces is shown for the whole data period in Fig. 4. Even though panel Fig. 4a shows only GWs classified as mountain waves, the

**Figure 2.** Observed GWMF at 35 km altitude (top row) for all sources and (bottom row) for mountain waves only for two periods. The periods are chosen as just before the SSW unfolds and after the vortex has recovered.

**Figure 3.** Timeline of observed GWMF at 35 km altitude north of 40°N during the unfolding of the SSW (SSW onset on 31st December, Rao et al. (2019)). The part of the observed GWMF that can be attributed to orographic sources is shown with hatching.

distribution does not fully resemble the underlying topography. The distribution is far spread, in particular above the northern






Atlantic. A direct linkage of the observations to topography below would result in misleading findings. For example, the GWMF to the east of Greenland would be missed, even though it is of orographic origin and propagated laterally up to this altitude. Furthermore, there is some GW activity above the Great Planes of the USA, where no orographic origin would be assumed at all due to the flat terrain if the lateral propagation is not accounted for.

The lateral propagation of the MWs is illustrated in Video Supplement S1, which shows the GW backward ray tracing from observation altitude (Fig. 4a) down to 5 km altitude (roughly corresponding to Fig. 4b). Snapshots of the video supplement and a brief description of key points are given in App. B. In some cases, the GWs propagate extremely far from their sources: some waves observed above Scandinavia can be traced back to Newfoundland. Even though these GWs might be assessed as MWs depending on their location at observation altitude, their source would be attribute to the wrong mountain ranges.

The distribution of GWMF carried by the MWs at or close to source level is shown in Fig. 4b (the location of each ray is taken at the lowermost altitude of the ray tracing data). A strong focusing onto mountain ranges due to the backward ray tracing can be seen and the distribution closely resembles the underlying topography of Fig. 4c. The individual mountain ranges are much better discernible than at observation altitude, and a few, e.g., the Brooks range in northern Alaska, become visible only after the backward ray tracing since most excited MWs propagate away from the source until the satellite would observe them.

Associating only the GWs observed directly above the mountain ranges with orographic origin would lead to missing important sources, such as the Brooks range, switch up source regions for far-propagating waves, and underestimate the orographic GWMF by roughly a half (as will be further detailed in the next section).

# 5.2 Oblique propagation distances

Figure 5 shows the fraction of GWs and absolute GWMF that has propagated more than the given horizontal distance from its source to the observation at 35 km altitude. It therefore gives an idea of how far the bulk of the MWs propagate horizontally from excitation at their source. This supports that the direct linking between observed GWMF in the middle atmosphere and possible sources directly below can not be reliably done. Around 60% of the observed GWs (accounting for about 45% of total GWMF) are observed farther than 1 000 km from their source and around 10% (roughly 7.5% of total GWMF) even farther than 5 000 km.

Comparing the number of GWs and the GWMF-weighted distribution shows that the latter has a steeper cumulative distribution function. From this, we can conclude that GWs carrying higher GWMF tend to propagate less far from their source horizontally. This is in agreement with GWs with long vertical wavelengths, which carry higher momentum (e.g., Ern et al., 2004), propagating more vertically than those with shorter vertical wavelengths.

The distribution of propagated distances versus altitude is shown in Fig. 5b. On average, the GWs propagate nearly linear with a slope of about 65 km horizontal per ascended vertical kilometer (as determined by a linear fit). The median distance is much smaller, which shows that the distribution of propagation distances is fairly skewed towards high values. Even though individual GWs show enhanced lateral propagation in the altitude range between about 8 and 15 km (see also Video Supplement S1 and App. B), the ensemble does not exhibit a clear altitude range of lateral propagation but rather a smooth horizontal propagation on average (although it show a slight bump compared to the linear fit).

**Figure 4.** Orographic GWMF distribution at observation altitude of 35 km (a) and at the lowest altitude to which the MWs could be traced (b). Note that the GWMF for each individual GW in panel b is the same as initialized in panel a. Panel c shows the surface elevation and the labels depict individual mountain ranges and regions: 1 - Brooks Range, 2 - Rocky Mountains, 3 - New England, 4 - Appalachians, 5 - southern Greenland, 6 - Iceland, 7 - British Isles, 8 - Spitsbergen, 9 - Scandinavian Alps, 10 - Alps, 11 - Ural, 12 - Novaya Zemlya, and 13 - Mongolia.

## 6 Conclusions

Analyzing and understanding observed and modeled GWs distributions is challenging, as their characteristics vary strongly with source processes and propagation conditions. In particular, the ability of GWs to travel long horizontal distances makes it difficult to attribute observed waves to their sources by mere collocation. Ray tracing offers a tool to disentangle different parts of the observations, especially for orographic GWs (mountain waves, MWs).

In this study, we combined simulated observations from a simulated space-based infrared imager (the ESA Earth Explorer 11 candidate CAIRT) with backward ray tracing to isolate the orographic part of the observed GWs. The MWs were identified



**Figure 5.** a) Cumulative distribution function of number of MWs (orange) and carried GWMF (teal) versus the horizontal propagation distance form their source. The distances for the 50th, 75th, and 90th percentile have been marked by dashed lines. Note the logarithmic horizontal axis. b) Propagation distance from the estimated source versus altitude. Shown is the mean (red), the median (black) and the 1st and 3rd quartiles (gray shading).

based on two criteria: a slow ground-based phase speed and backward trajectories terminating close to the surface and above land. To assess the robustness of this approach, we also tested the sensitivity of backward ray tracing to uncertainties in horizontal and vertical wavelengths as well as wave directions caused directly by measurement uncertainties or uncertainties in the spectral analysis(see App. A).

Our analysis focuses on the northern hemisphere in the period of 2018-12-10 to 2019-03-29, which covers the 2018/2019 New Year sudden stratospheric warming (SSW). We showed that the orographic part of the observed GWs can be reliably isolated. As expected, MW activity was concentrated above major mountain regions, including Mongolia, the Himalayas, Greenland, Iceland, Scandinavia, and northern Canada. Furthermore, MW activity was highly variable in time: while on average only about one-sixth of the GW momentum flux (GWMF) at 35 km altitude originated from orographic sources, this fraction occasionally increased to nearly one-half. A few days before the onset of the SSW, enhanced and persistent GWMF was traced back to Mongolia, whereas after the vortex recovery, the North Atlantic and northern Canada became the dominant regions for MWs. These results suggest a possible connection between anomalously elevated GWMF over Mongolia and the onset of the SSW, though the causal relationship remains uncertain.

An aggregated view further confirmed the robustness of our MW-identification approach. The spatial distribution of MW termination points closely mirrored surface topography, confirming the conclusion that the isolated waves were indeed of orographic origin. The analysis also revealed long-distance propagation: more than half of the MWs in our simulation were observed more than 1 300 km from their source, accounting for about 38% of the total GWMF observed at 35 km altitude. This demonstrates that simple collocation of GW observations with topography can be highly misleading, which can be seen also in Video Supplement S1 and App.B.





The methodology can, in principle, be applied to any data set (observation or model) providing 3D data on global (or part-global) scale. The effectiveness of our methodology depends strongly on the accuracy of the spectral analysis used to derive GW parameters. Provided these can be determined with sufficient precision, the approach enables robust isolation of orographic waves and opens new opportunities to investigate their role in preconditioning the polar vortex prior to SSW events. Future applications could extend the method to other GW sources, such as convection, by comparing ray trajectories with regions of precipitation and latent heat release.

*Data availability.* The simulated temperature observations are openly available on Zenodo (DOI: 10.5281/zenodo.17251039). The ray tracing data and the S3D spectral analysis data used in this study are available upon request.

210 Video supplement. Video Supplement S1 is available on Zenodo(DOI: 10.5281/zenodo.17275660).

## Appendix A: Ray tracing sensitivity on wave parameter uncertainties

Ray tracing requires the knowledge of the 3D wave vector of the GW, i.e., horizontal and vertical wavelength as well as the (horizontal) wave direction have to be known with high certainty. If this is not given, the calculated trajectories of the analyzed GWs are not reliable. In this section, we estimate the sensitivity of backward ray tracing using a situation on 2019-03-19, where orographic GWs observable above Scotland at 35 km altitude can be backward ray traced to Iceland. Note that we consider all GWs detected in the full model data here instead of simulated satellite observations.

Figure A1a and b show the initial GWMF distribution at 35 km altitude and at termination altitude as ray traced directly from the model GWs without any perturbations to the wave parameters. As can be seen, many GWs can be backtraced to Iceland and the southern coast of Greenland. The distribution is much more localized than at initialization altitude and we can assume that both, Iceland and southern Greenland, are major orographic GW sources in this situation. In the following, we will investigate if this can still be done if the GW parameters are perturbed due to, e.g., measurement noise or uncertainties in the spectral analysis.

Figure A1c-h show the resulting backtracing distributions, where the individual parameters of all initialized waves have been perturbed by fixed  $\pm 10\%$  (in case of horizontal and vertical wavelengths) or  $\pm 10^\circ$  (in case of the wave direction). Comparing Fig. A1c and d to the reference in Fig. A1b, we see that the GW trajectories are not too sensitive on the horizontal wavelength. The source hotspots are still well visible even though the focus point above Iceland shifts slightly to east or west with reduced and increased horizontal wavelength, respectively. The vertical wavelengths, on the other hand, have a much stronger impact on the GWMF distribution (Fig. A1e and f). In particular, the GWs supposedly originating form Iceland trace back much further to the west (east) for reduced (increased) vertical wavelength leading to the hypothesis of these waves originating there becoming implausible.


Figure A1. Backward ray tracing from the model data of 2019-03-19 at initialization altitude of 35 km (a) and termination (b-h) in terms of GWMF. Panel b shows the reference termination locations, while the respective wave parameter has been perturbed by -10% (left column) or by +10%/° (right column).

The effect of the wave direction (Fig. A1g and h) is different in the sense that the focusing of GWs that is seen in the reference (Fig. A1b) is no longer as pronounced. The resulting GWMF distributions spread farther around Iceland than in the previous cases. The source assessment will also suffer for this reason and, possibly, less GWMF would be attributed to orographic origin.

As we can see, all wave parameters are important for reliable ray tracing, however, the horizontal wavelength does not contribute to uncertainties as much as the other parameters. Most important is a reliable and accurate estimation of the wave direction and vertical wavelength.

We further quantify the acceptable uncertainties in the wave parameters via Monte Carlo simulations: the three wave parameters of all waves are perturbed using Gaussian noise of a fixed standard deviations in percent (for the wavelengths) and



**Figure A2.** Violin plots for the horizontal distances from the reference backward ray tracing termination location for different error setups. In each case, the vertical and horizontal wavelength and the wave direction have been perturbed with Gaussian error of A) 3%/°, B) 5%/°, C) 10%/°, and D) 15%/°. Panel a shows absolute distances, panel b distance in horizontal wavelengths. Horizontal blue lines depict the 1st quartile, median, and 3rd quartile.

degrees (for the wave direction). 100 instances are initialized and backward traced for four standard deviations of 3, 5, 10, and 15%/° (in the following referred to as cases A, B, C, and D, respectively). The horizontal distance of the termination location of each ray in the perturbed case compared to the reference unperturbed ray tracing serves as the measure of reliability.

The distributions of horizontal distance of the ray traces compared to their reference location for the four different perturbation cases are shown in Fig. A2. The low level of uncertainty in cases A and B lead to very low spread from the reference: More than 75% of the rays trace within 200 km and 300 km of the reference location, respectively. In both cases, more than 80% of GWs are within one wavelength of their reference backward ray trace. With case C (i.e., an uncertainty of 10%/°), the backward ray tracing starts to spread much farther: Now, only about 60% of waves trace back within 300 km of the reference location and about the same within one wavelength. The tail of the distribution towards longer distances becomes much more pronounced. Finally, in case D, less than half of the waves can be traced back within one wavelength of their reference location. Moreover, the tail becomes even longer with the 3rd quartile located around 1 000 km or 3 wavelengths.

To associate a wave with an underlying source, and hence get reliable backward tracing results, a mismatch with the actual location of about one wavelength is reasonable. More than that and it will be increasingly difficult to match source regions with the GW trajectories. Therefore, we estimate that case C with a (Gaussian) uncertainty of 10%/° is a reasonable edge case, where ray tracing is still meaningful and the back traced locations can be trusted.



## 255 Appendix B: Oblique propagation of MWs

Video Supplement S1 shows the ray tracing results in form of horizontal maps from observation altitude at 35 km down to 5 km. Figure B1 shows three snapshots of the video supplement at 30, 20, and 10 km altitude. From higher to lower altitudes, the GWMF distribution increasingly focuses on the main mountain range features such as Scandinavia and coastal Greenland. This focusing is also seen in northern Alaska, where GWMF appears above the Brooks range at 10 km altitude while it was invisible at observation altitude of 35 km (Fig. 4a). In this case, the source only becomes apparent once the backward trajectories are analyzed.

Furthermore, Video Supplement S1 also shows the oblique propagation of the MWs. In Fig. B1, two waves are highlighted by the black and gray arrows, respectively. The black arrows point to a MW carrying high GWMF that can be traced from its observation above Scandinavia all the way back to Newfoundland. A simple collocation with sources below might have attributed it to orography as well but still completely missing the actual source region. Similarly the MW highlighted by the gray arrow is also observed eastward of Scandinavia but can be traced back to Iceland. In this instance, the MW propagates very strongly in the horizontal below around 15 km. This behaviour of a rather narrow "horizontal propagation layer" is not uncommon for the investigated GWs.

Author contributions. SR performed the development of the methodology, analysis of data, and writing of the paper.

Competing interests. The author declares that there are no competing interests.

Acknowledgements. Special thanks to Peter Preusse and Jörn Ungermann for the helpful discussions and providing the simulated retrieval of the satellite data.

*Financial support.* This study has been funded by the European Space Agency (ESA) via PerReC (CAIRT Phase A Mission Performance and Requirement Consolidation) under ESA contract no. 4000136480/21/NL/LF.

**Figure B1.** Snapshots of Video Supplement S1. Panels a, b, and c show the GWMF distributions as determined from ray tracing at 30, 20, and 10 km (cf. position bar on the right), respectively. Each GW is weighted by its GWMF at observation altitude.

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
