# Peer review of "Identifying Orographic Gravity Waves in 3D Observations via Backward Ray Tracing"

_EGUsphere, 2025_

## Author Comment (AC1)

**1 Response Referee No. 1**

We thank the reviewer for the thorough review of our article. The comments and suggestions were very helpful in improving the presentation, quality, and relevance of our manuscript.

The responses to specific comments are given below. The original reviewer comments are given in italic and any text given in blue has been added to the manuscript in response to the comment.

**1.1 Major comments**

1. *Propagation: Once a mountain wave is excited at the surface, the further propagation is determined by the wind field. This issue deserves some discussion and presentation with typical examples. The key process is likely the refraction into the stratospheric jet, and this should be identified in the data. Further, there are comparable field campaigns which are documenting such oblique propagation in detail - and these should be discussed.*

   A brief description of the campaigns targeting oblique propagation of MWs and references for further deep diving are now provided in the introduction:

   In the case of mountain waves, the oblique propagation was the target of previous field campaigns such as the DEEPWAVE [Fritts et al., 2016, Eckermann et al., 2016, Portele et al., 2018], POLSTRACC/GW-Lcycle [Krisch et al., 2017, Geldenhuys et al., 2021], and SouthTRAC [Rapp et al., 2020] campaigns and the identification of sources was demonstrated from limb-imager observations from research aircraft [Krisch et al., 2020, Krasauskas et al., 2023].

   Regarding the interactions of GWs and the polar vortex and its importance for the refraction and propagation, we added a section discussing the wind situation during th SSW - see reply to the next comment.

2. *Stratwarms: The wind field is changing drastically during sudden stratospheric warmings, which concerns the phases before, at, and after the central date. The specific spatial structure of the wind should be documented and used to argue for the local gravity wave appearance above Mongolia respectively Atlantic/Canadian.*

   This reply addresses both above comments:

   In regards to the wind profiles and situation before, during and past the SSW, a new section has been included in the paper:

   The GWMF distribution in Fig. 2 and the time series in Fig. R2 can be better understood by investigating the wind situation during periods. Figure R1a-c shows the strength and location of the polar vortex in the northern hemisphere at 35 km altitude averaged over 2 weeks each. Just before the SSW, the vortex was already displaced southward towards Europe and Asia. MWs excited in Mongolia encounter thus an ideal situation for propagation towards the center of the polar vortex. In particular, the far southward extent, in combination with favorable surface winds, is one reason why such strong orographic GW activity is observed in this region.

   During the SSW (Fig. R1b), in contrast, the vortex ceases to exist, leading to critical level filtering of MWs and GWs of other origins. The wind reversal can be seen more clearly in Fig. R1d. At the considered altitude of 35 km, the wind reverses around December 26th, which directly correlates with the lack of observed GWMF in Fig. R2a (pink arrow).

   After the SSW, the vortex recovers (Fig. R1c) but does not extend as far southward as before above Asia, providing worse propagation conditions for MWs. In particular, the wind directly above Mongolia and the Himalayas is almost zero, leading to strong filtering. On the other hand, the vortex consolidates above the Northern Atlantic, which, in combination with the governing favorable low-level winds (not shown), leads to high amounts of GWMF observed in this region.

**1.2 Technical comments**

1. *L10: You write that orographic gravity waves may "dominate episodically, including prior to the onset of SSW" - but you do not show it. In line 132 you write of "40 % on individual days" but this is far from dominance. Please, reformulate this passage to the features you are documenting with the analysis.*

   Indeed this was maybe a bit to euphemistic. The referenced sentence was changed to "but episodically they can account for a major part of the GWMF, including prior to the onset of the SSW". Furthermore, other references have been changed as follows:
   "before the SSW, the orographic GW activity is strongly localized above Mongolia,..."
   "Canada completely dominate in terms of orographic GWMF after the vortex recovered."

[Figure]

Figure R1: Wind situation during the SSW. Panels a-c show two-week averaged zonal wind speed at 35 km altitude before, during, and after the SSW. Panel d shows the time-altitude cross section of zonal mean zonal wind at 60°N.

2. *L69: You write the "spectra are smoothed" - so, you did not execute this filtering in space? Further, the cutoff at zonal wavenumber 7 ( 2900 km at 60 °N) and 10° in latitude ( 1100 km) is not consistent. At least, this is not orientation preserving. Some arguments for this procedure are given in Mathew et al. (2025), which I after a while found at*

   *https://egusphere.copernicus.org/preprints/2025/egusphere-2025-4602/*

   *(a DOI in the reference list would have been helpful). These are more technical reasons like strong jumps and applicability of smoothers. Please, adjust the text accordingly.*

   First, I apologize for not finding the article easier. On submission, Mathew et al. [2025] was accepted for preprint publication but had no DOI yet.

   Indeed, the filtering is done in spectral space, i.e. zonal FFT. The smoothing is then performed in space by smoothing the PW spectra (complex amplitudes) in the latitude. We would love to extend the zonal wavenumber to higher values for the scale separation but it is limited by the number of orbits per day of the satellite instrument. In the meridional direction, we are more flexible.

   The text has been updated to better reflect the technical limit: "Note that the rather low cutoff at ZWN 7 is used, as this would also be an applicable cutoff for CAIRT observations [Mathew et al., 2025]. This number is limited by the orbits per day of the satellite."

3. *L119: I suggest to write "before and past" and leave "way" out.*

   Changed as suggested.

4. *L126: Do you mean "time series" with "timeline"?*

   Indeed, "time series" is the better wording here. Changed accordingly.

5. *L131: I do not see that the strongest GW events are "mostly driven by orographic GWMF" - please, reformulate.*

   The text has been reformulated to better reflect what the data shows: "Nevertheless, days with anomalously high GWMF often correlate with strong orographic GWMF, indicating that MWs play a major role in these high GWMF events."

6. *Fig. 2: Please, specify what the contours are. In view of the major comments, I suggest to indicate the horizonal wind speed contours in order to show the refraction effect.*

   Missing the description of the contours was a clear oversight. They are indeed the horizontal wind speeds. The caption has been changed accordingly:

   "Observed GWMF at 35 km altitude (top row) for all sources and (bottom row) for mountain waves only for two periods. Contours show the horizontal wind speed, i.e., the location of the polar vortex. The values of the contours are chosen as 25, 50, and 75 ms$^{-1}$, respectively. The periods are chosen as just before the SSW unfolds and after the vortex has recovered."

7. *Fig. 3: Do you mean "time series" with "timeline"? Perhaps, indication of the central date with an arrow would help identification of the special situation. So, what you show here is a polar cap averaged of absolute momentum flux, right? Is it possible to say something on the sign which could change when the zonal wind turns easterly?*

   This also meant to be the "time series". The wording was changed and, as you suggested, a panel showing only the zonal GWMF added. And indeed, the direction of the GWMF changes during the SSW. The following text has been added:

   Figure R2b shows the zonal GWMF separated for eastward and westward direction. As expected, the orographic GWs are directed mainly westward with some meridional component during the SSW (seen via the lack of zonal GWMF compared to the total GWMF in Fig. R2a). As the SSW unfolds and the vortex reverses, the net zonal GWMF changes from westward to eastward for brief periods around December 26th and again around January 6th for around 10 days.

[Figure]

Figure R2: Time series of observed GWMF at 35 km altitude north of 40°N during the unfolding of the SSW (SSW onset on 31st December, Rao et al. [2019]). Panel a) shows the total absolute GWMF, panel b) the zonal GWMF summed for eastward and westward directions separately. The part of the observed GWMF that can be attributed to orographic sources is shown with hatching in both panels. The pink and green arrows point at the time of the wind reversal at 35 km and the central SSW date, respectively.

8. *L175: "it show" –> "it shows"*

   Changed as suggested.

9. *Fig. 4: Also here, overplots of wind speed could help interpretation. From which time are these plots, or are they averages?*

   Fig. 4 shows an accumulated view on the observation period to show the validity of the approach with as many data points as possible. The following sentence has been added to the caption for clarification: Here, the full observation period of December 10th to March 31st is accumulated.

10. *L181: This sentence is a bit confusing, I guess you mean "simulated observations of a space-based infrared imager", or?*

    Thanks for pointing out the unclear wording. The wording was changed as suggested.

11. *Fig. 5: May be, another horizontal line in the left plot for 0.1 fraction would visualize the 90th percentile*

    Changed as suggested.

12. *L183: With reference to L108ff, I see three instead of two criteria.*

    Yes indeed, three is more compatible with the rest of the paper. The thought here was that one criterion is the slow phase speed and the other backward ray tracing (consisting of two checks). Changed as you suggested.

13. *Fig. A1: Please, specify "orographic GWMF" as you did for the other figures.*

    App. A showcases the general sensitivity of the backward ray tracing on the different parameters as a prerequisite for the methodology described in the main text. Hence, all detected GWs are shown in Fig. A1 for the full picture. A clarification has been added to the caption of Fig. A1: Note that this

[Figure]

Figure R3: Averaged wind profiles along the GW paths (panel a) and cumulative propagated distance (panel b). The data was split by propagation distance: the blue (solid) line shows GWs that have already propagated further than 800 km from their source at 20 km altitude. The red (dashed) line shows the respective complement. Shaded regions give the 1st and 3rd quartile, respectively.

shows all observed GWMF (not only the orographic part) as a showcase of ray tracing sensitivity on the different parameters.

14. *L254: May be, "limit case" is better to read than "edge case".*

    Changed as suggested.

15. *L264: The mentioning of the "horizontal propagation layer" is interesting - is it the tropospheric jet or the lower edge of he stratospheric jet? A further documentation of wind profiles could make this point clearer and worth to be placed in the main text.*

    From the investigations in this study, the reason for the localized propagation of some GWs can be traced to a better separation of tropospheric and stratospheric jet, i.e., a more pronounced wind minimum around 230 km. The resulting wind shear leads to stronger refraction and subsequently oblique propagation of the GWs. This is an interesting field of research and we believe it would be worth a more rigorous study than possible in this work. Therefore, the topic is only extended in the appendix and not moved to the main text:

    One reason for this layer, where some GWs propagate anomalously far horizontally can be seen in Fig. R3. The far propagating GWs encounter stronger wind shear caused by a better separation of the tropospheric and stratospheric jet. The local wind minimum leads to the refraction of GWs, which in turn leads to stronger oblique propagation.

**References**

S. D. Eckermann, D. Broutman, J. Ma, J. D. Doyle, P.-D. Pautet, M. J. Taylor, K. Bossert, B. P. Williams, D. C. Fritts, and R. B. Smith. Dynamics of orographic gravity waves observed in the mesosphere over the Auckland Islands during the deep propagating gravity wave experiment (DEEPWAVE). *J. Atmos. Sci.*, 73 (10):3855–3876, OCT 2016. ISSN 0022-4928. doi: 10.1175/JAS-D-16-0059.1.

D. C. Fritts, R. B. Smith, M. J. Taylor, J. D. Doyle, S. D. Eckermann, A. Doernbrack, M. Rapp, B. P. Williams, P. D. Pautet, K. Bossert, N. R. Criddle, C. A. Reynolds, P. A. Reinecke, M. Uddstrom, M. J. Revell, R. Turner, B. Kaifler, J. S. Wagner, T. Mixa, C. G. Kruse, A. D. Nugent, C. D. Watson, S. Gisinger, S. M. Smith, R. S. Lieberman, B. Laughman, J. J. Moore, W. O. Brown, J. A. Haggerty, A. Rockwell, G. J. Stossmeister, S. F. Williams, G. Hernandez, D. J. Murphy, A. R. Klekociuk, I. M. Reid, and J. Ma. The deep propagating gravity wave experiment (DEEPWAVE): An airborne and ground-based exploration of gravity wave propagation and effects from their sources throughout the lower and middle atmosphere. *Bull. Amer. Meteor. Soc.*, 97(3):425–453, MAR 2016. ISSN 0003-0007. doi: 10.1175/BAMS-D-14-00269.1.

M. Geldenhuys, P. Preusse, I. Krisch, C. Zülicke, J. Ungermann, M. Ern, F. Friedl-Vallon, and M. Riese. Orographically induced spontaneous imbalance within the jet causing a large-scale gravity wave event. *Atmos. Chem. Phys.*, 2021. doi: 10.5194/acp-21-10393-2021.

L. Krasauskas, B. Kaifler, S. Rhode, J. Ungermann, W. Woiwode, and P. Preusse. Oblique propagation and refraction of gravity waves over the Andes observed by GLORIA and ALIMA during the SouthTRAC campaign. *J. Geophys. Res. Atmos.*, page e2022JD037798, 2023. doi: 10.1029/2022JD037798.

I. Krisch, P. Preusse, J. Ungermann, A. Dörnbrack, S. D. Eckermann, M. Ern, F. Friedl-Vallon, M. Kaufmann, H. Oelhaf, M. Rapp, C. Strube, and M. Riese. First tomographic observations of gravity waves by the infrared limb imager GLORIA. *Atmos. Chem. Phys.*, 17(24):14937–14953, 2017. doi: 10.5194/acp-17-14937-2017.

I. Krisch, M. Ern, L. Hoffmann, P. Preusse, C. Strube, J. Ungermann, W. Woiwode, and M. Riese. Superposition of gravity waves with different propagation characteristics observed by airborne and space-borne infrared sounders. *Atmos. Chem. Phys.*, 20(19):11469–11490, 2020. doi: 10.5194/acp-20-11469-2020. URL https://acp.copernicus.org/articles/20/11469/2020/.

A. J. Mathew, S. Rhode, M. Ern, M. Pramitha, and P. Preusse. GLOFI – A methodology and toolbox for scale-separation of satellite observations for analysis of gravity waves. *EGUsphere*, 2025:1–23, 2025. doi: 10.5194/egusphere-2025-4602. URL https://egusphere.copernicus.org/preprints/2025/egusphere-2025-4602/.

T. C. Portele, A. Doernbrack, J. S. Wagner, S. Gisinger, B. Ehard, P.-D. Pautet, and M. Rapp. Mountain-wave propagation under transient tropospheric forcing: A DEEPWAVE case study. *Mon. Weath. Rev.*, 146(6): 1861–1888, JUN 2018. ISSN 0027-0644. doi: 10.1175/MWR-D-17-0080.1.

J. Rao, C. I. Garfinkel, H. Chen, and I. P. White. The 2019 new year stratospheric sudden warming and its real-time predictions in multiple s2s models. *J. Geophys. Res. Atmos.*, 124(21):11155–11174, 2019. doi: https://doi.org/10.1029/2019JD030826. URL https://agupubs.onlinelibrary.wiley.com/doi/abs/10.1029/2019JD030826.

M. Rapp, B. Kaifler, A. Dörnbrack, S. Gisinger, T. Mixa, R. Reichert, N. Kaifler, S. Knobloch, R. Ecbert, N. Wildmann, A. Giez, L. Krasauskas, P. Preusse, M. Geldenhuys, M. Riese, W. Woiwode, F. Friedl-Vallon, B. Sinnhuber, A. de la Torre, P. Alexander, J. Hormaechea, D. Janches, M. Garhammer, J. Chau, F. Conte, P. Hoor, and A. Engel. Southtrac-gw: An airborne field campaign to explore gravity wave dynamics at the world's strongest hotspot. *Bull. Amer. Meteor. Soc.*, 2020. submitted.